# THE UNREASONABLE INEFFECTIVENESS OF THE DEEPER LAYERS

**Andrey Gromov**[*]
Meta FAIR & UMD

**Kushal Tirumala**[*]
Meta FAIR

**Hassan Shapourian**
Cisco

**Paolo Glorioso**
Zyphra

**Daniel A. Roberts**
MIT & Sequoia Capital

## ABSTRACT

How is knowledge stored in an LLM's weights? We study this via layer pruning: if removing a certain layer does not affect model performance in common question-answering benchmarks, then the weights in that layer are not necessary for storing the knowledge needed to answer those questions. To find these unnecessary parameters, we identify the optimal block of layers to prune by considering similarity across layers; then, to "heal" the damage, we perform a small amount of finetuning. Surprisingly, with this method we find minimal degradation of performance until after a large fraction (up to half) of the layers are removed for some common open-weight models. From a scientific perspective, the robustness of these LLMs to the deletion of layers implies either that current pretraining methods are not properly leveraging the parameters in the deeper layers of the network or that the shallow layers play a critical role in storing knowledge. For our study, we use parameter-efficient finetuning (PEFT) methods, specifically quantization and Low Rank Adapters (QLoRA), such that each of our experiments can be performed on a single 40GB A100 GPU.

## 1  INTRODUCTION

In this work we study a very simple pruning strategy using open-weight LLMs. In particular, we develop a method that uses the similarity between the representations at different layers to identify the optimal layers to prune for a given pruning fraction; then, after removing these layers we "heal" the pruning-induced mismatch with a small amount of fine tuning (using QLoRA). Our main result is that we can remove a substantial fraction of the *deepest layers* from models with minimal degradation in downstream question-answering benchmarks. For example, for Llama-2-70B (Touvron et al., 2023) we can eliminate up to roughly *half* of the layers before the performance collapses. An overview of our strategy and the results of pruning Llama-2-70B are shown in Figure 1.

Our intuition for dropping layers comes from considering the residual structure of the transformer architecture. In more detail, the output of the final layer can be decomposed as a sum over the outputs of all the model layers plus the embedded input. If such a sum had numerous and independent terms, then removing a handful of them should not significantly change the output. However, since the terms are not independent – each layer is input to the following layer – we should expect to be able to remove terms if the residual contribution from a particular layer is small. In other words, if the output of each layer does not change too much from layer to layer.[1]

---

[*]Co-first authors; please direct correspondence to the union of {gromovand@meta.com, kushaltirumala99@gmail.com, drob@mit.edu}.

[1]This is strongly suggested by "lens" investigations that studied the evolution of the token distribution as a function of layer index such as the "logit lens" (nostalgebraist, 2020) and the "tuned lens" (Belrose et al., 2023). A separate line of reasoning along these lines previously inspired neural ODEs (Chen et al., 2018), and led Yang et al. (2023) to argue that ideally representation should change substantially from layer to layer in order to most effectively make use of the parameters of a network.

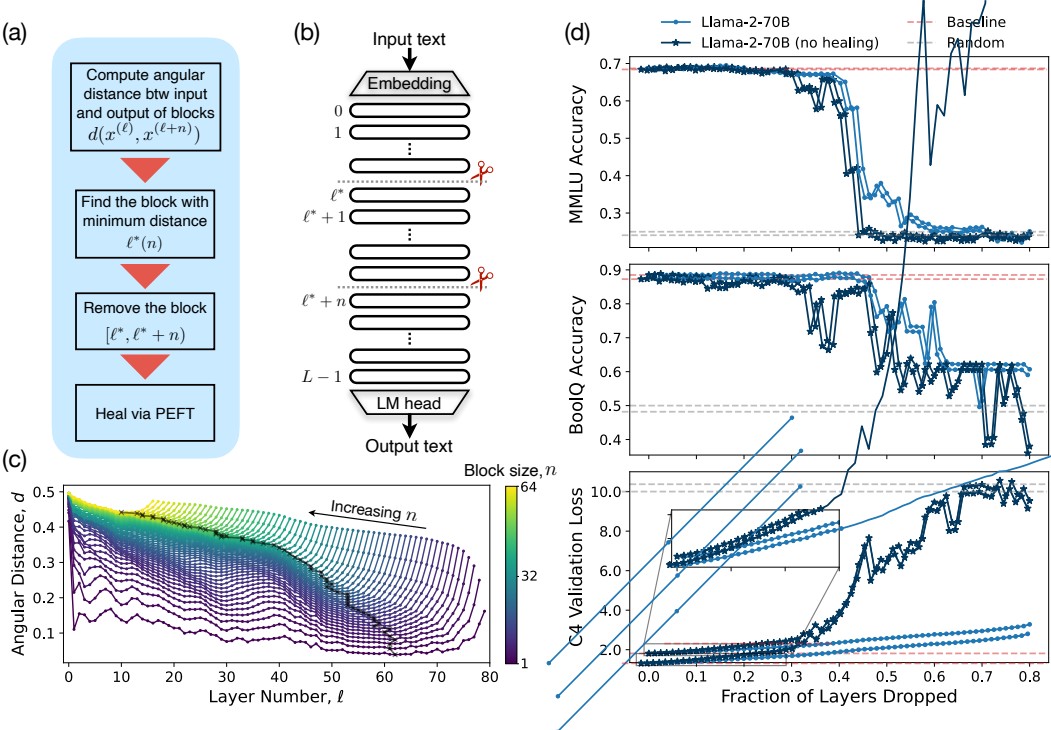

Figure 1: Overview of our layer-pruning strategy and example results: *(a)* a flowchart describing the algorithm: if removing $n$ layers, we find the layer, $\ell^*$, that minimizes the angular distance, $d$, between layers $\ell$ and $\ell+n$; we then remove the $n$ layers beginning with layer $\ell^*$; finally, if necessary, we can "heal" the damage with a small amount of (parameter-efficient) finetuning. *(b)* a schematic depicting the removal of $n$ total layers, indexed from $\ell^*$ to $\ell^*+n-1$. *(c)* angular distance, $d$, between different numbers of layers, $n$, vs. the layer number, $\ell$, that indexes the beginning of the block of $n$; the bottom curve (darkest purple) represents $n=1$, while the top curve (lightest yellow) represents $n=64$; the black line traces $\ell^*(n)$, the minimum of the angular distance across the different sized layer blocks. *(d)* results of pruning Llama-2-70B with healing (light blue) and without healing (dark blue) as a function of the fraction of layers removed: the top (middle) panel gives the accuracy on the MMLU (BoolQ) question-answering benchmark, while the bottom panel the autoregressive loss on a subset of the C4 validation set; here, the dashed red lines (dashed gray lines) indicate the accuracy or loss of the original unpruned model (of random guessing); these plots illustrate that typical behavior we find in which there are sharp transitions in performance for the accuracy of question-answering tasks (here between 40%-50% pruning fraction), but continuity and very slow growth in the healed loss (light blue) up to at least to 80% pruning fraction.

In conjunction with our layer pruning, we investigate the similarity of layer representations at different separations and find broadly that deeper layers are qualitatively more similar to neighboring layers than shallow layers (with the exception of the very final layer). This suggests an even simpler pruning strategy: remove layers beginning at the penultimate layer and proceed from deep to shallow until the desired number of layers have been removed. In this case, we find that, after healing the damage with a small amount of QLoRA finetuning, we can achieve performance that nearly matches the more involved similarity-informed layer pruning strategy. The effectiveness of this method is evidence that LLMs might not properly leverage the parameters in the deeper layers of the network.

That said, while question-answering (QA) benchmarks such as MMLU and BoolQ are robust to a large amount of layer pruning, other measures of performance are not: if we look at the loss on next-token predictions for an IID dataset (C4 validation set), we find that the model is smoothly damaged in proportion to the fraction of the number of layers pruned. Since perplexity typically correlates strongly with downstream metrics, this naturally begs the question: which tasks are less robust than QA benchmarks to pruning? As part of our final discussion, we explore reasoning

related tasks (GSM8k and HellaSwag) and see that they are harmed by any amount of pruning. Altogether, this leads to the following accounting of state: the shallow layers likely play a critical role in the storing of knowledge and retrieving of information, while the deeper layers are important for higher-level computations such as mathematical reasoning.

The structure of this paper is as follows. In §2, we first perform a literature review of both practical post-training strategies and science-of-deep-learning investigations that motivate our work. Then, in §3, we give intuition for our layer pruning strategy and explain our method in detail, while in §4 we iterate over all our experimental results. Finally, we conclude in §5 by exploring tasks beyond QA benchmarks, such as reasoning, and highlighting directions of future work. Specific model, finetuning, dataset, and evaluation details can be found in Appendix B, and evaluation ablations can be found in Appendix C.

## 2    LITERATURE REVIEW

Pruning for neural networks has a long history (LeCun et al., 1989; Hassibi and Stork, 1992): while initial work focused on *unstructured pruning* (Han et al., 2015; Chen et al., 2015; Srinivas and Babu, 2015), *structured pruning* techniques were developed to make sparse networks more efficient (Li et al., 2016; Wen et al., 2016; Hu et al., 2016; He et al., 2017; Huang et al., 2018; Murray and Chiang, 2015; See et al., 2016; Kim and Rush, 2016). Recent work, of course, focused on structured pruning of transformers (Voita et al., 2019; Michel et al., 2019; Kim and Awadalla, 2020; Fan et al., 2019; Zhang and He, 2020; Fan et al., 2021; Jha et al., 2023; Sajjad et al., 2023; Liu et al., 2023a; Hou et al., 2020; Sharma et al., 2023; Ashkboos et al., 2024; Xia et al., 2022; Lagunas et al., 2021; Men et al., 2024). Our work focuses on pruning the layers of decoder-only GPT style open-weight *large* language models after they've been pretrained. For an extended literature review, please see Appendix A.

## 3    METHOD

In this section, we give intuition for why we think layer pruning works (§3.1) and then we explain our method in detail (§3.2).

### 3.1    INTUITION

Our intuition for layer dropping comes from thinking about the representations as a slowly changing function of layer index. In particular, the layer-to-layer evolution of representations for a transformer is given by a *residual* iteration equation

$$x^{(\ell+1)} = x^{(\ell)} + f(x^{(\ell)}, \theta^{(\ell)}), \tag{1}$$

where $(x^{(\ell)}, \theta^{(\ell)})$, respectively, are the multi-dimensional input and parameter vectors for layer $\ell$, and $f(x, \theta)$ describes the transformation of one multi-head self-attention *and* MLP layer block. As for any residual network, if we unroll this iteration, we see that after $L$ total layers the output is described as a sum over the transformations of all the layers

$$x^{(L)} = x^{(0)} + \sum_{\ell=0}^{L-1} f(x^{(\ell)}, \theta^{(\ell)}). \tag{2}$$

If the terms in the sum were *numerous*, ($L \gg 1$), and *independent*, e.g. if the block functions were instead a function of the overall input as $f(x^{(0)}, \theta^{(\ell)})$, then perhaps any particular contribution to the sum (2) could be neglected.

Of course, they are not at all independent: if we delete layer $\ell - 1$, then we must now connect the old input to that layer, $x^{(\ell-1)}$, into the block function of layer $\ell$ as

$$x^{(\ell+1)} = x^{(\ell-1)} + f(x^{(\ell-1)}, \theta^{(\ell)}), \tag{3}$$

where, for clarity, we are not relabeling layers or inputs despite the deletion. In general, such a *mismatch* between the original input and new input should be very damaging for the network.

However, if, after some number of initial layers, the representations converge to a slowly changing function with respect to layer index,

$$x^{(\ell)} \approx x^{(\ell-1)} + \epsilon \,, \tag{4}$$

with $\epsilon \ll x^{(\ell)}$ in some appropriate sense, then the effect of deleting a particular layer $\ell$, e.g. making the replacement $x^{(\ell)} \to x^{(\ell-1)}$ in going from (1) to (3), should only change the representation in the subsequent layer, $x^{(\ell+1)}$, by a small amount. Similarly, to successfully prune the $n$ layers before layer $\ell$, i.e. those indexed from $\ell - n, \ldots, \ell - 1$, we'd want that the input to the pruned block should be very similar to the output of the pruned block:

$$x^{(\ell)} \approx x^{(\ell-n)} + \epsilon \,. \tag{5}$$

Regardless, any layer removal has a cascading effect: since post pruning $x^{(\ell+1)}$ is computed by a different function than before, cf. (1) vs. (3), and since then $x^{(\ell+1)}$ is directly or indirectly input to subsequent layers, $\ell + 2, \ldots, L$, deleting a shallow layer should have a much greater impact than deleting a deeper layer.

From this, we have the following hypotheses that we will test experimentally:

- *(0)* We should be able to prune layers of a residual network.
- *(1)* We should have greater success pruning deeper layers.
- *(2)* Blocks of layers we successfully prune should have outputs that are similar to their inputs.

In the next subsection, §3.2 we will explain the details of our pruning algorithm and in the following section, §4, we will present experimental evidence for points *(0)-(2)*.

### 3.2 LAYER-PRUNING ALGORITHM(S)

Our principal layer pruning algorithm is very simple:

0. Pick a a number of layers to prune $n$.
1. Compute the angular distance $d(x^{(\ell)}, x^{(\ell+n)})$, cf. (7) below, between the input to layer $\ell$ and the input to layer $\ell + n$ on a neutral pretraining dataset or on a dataset representative of a downstream task of interest.
2. Find the layer, $\ell^*$, that minimizes that distance:

$$\ell^\star(n) \equiv \operatorname*{arg\,min}_\ell \ d(x^{(\ell)}, x^{(\ell+n)}) \,. \tag{6}$$

3. Drop layers $\ell^\star$ to $\ell^\star+n-1$; connect the old input to layer $\ell^\star$ to the old $(\ell^\star+n)$th layer block.[2]
4. (Optionally) heal the mismatch at layer $\ell^\star + n$ with a small amount of fine tuning on a neutral pretraining dataset or particular dataset of interest.

If fewer words inside of a figure are more helpful to you than the text in an enumerated list, then note that this algorithm is also depicted in panels (a)-(b) of Figure 1.

Elaborating on the first step, the angular distance on a single sequence of length $T$ is given by

$$d(x^{(\ell)}, x^{(\ell+n)}) \equiv \frac{1}{\pi} \arccos\left( \frac{x_T^{(\ell)} \cdot x_T^{(\ell+n)}}{\left\| x_T^{(\ell)} \right\| \left\| x_T^{(\ell+n)} \right\|} \right) \,, \tag{7}$$

where the inner product is over the hidden dimension of the model for the final token $T$ of the sequence, $\| \cdot \|$ denotes the $L^2$-norm, and the factor of $1/\pi$ is a convention.[3] This distance should

---

[2]Layers are often contained in a data structure, such a `ModuleList` in *PyTorch*, so to drop these layers we would simply define a new `ModuleList` that removes the layers from $\ell^\star$ to $\ell^\star + n - 1$.

[3]Two comments: *(i)*, we do not expect our choice of angular distance – in lieu of any other reasonable metric, e.g., such as cosine similarity – to be particular significant; and *(ii)*, we chose to focus on the final token since, due to the causal attention mask, its embedding is the only one that depends on the entire sequence.

then be summed over a number of examples that is large enough to get a low-fluctuation estimate but overall should be quite small.

Elaborating on the "optionality" of the final step, we find that the near-lack of performance degradation on question-answering benchmarks, cf. Figure 1(d) and others in §4.1, can be extended to greater pruning fractions with a small amount of finetuning. Depending on resource constraints and intended application of the pruned model, this may not be necessary. However, the healing procedure does have a substantial impact on perplexity, cf. Figure 1(d) and others in §4.2.

For both the angular distance measuring and the healing, if the ultimate goal is to supervise finetune (SFT) a model for a downstream task, it could be useful to evaluate the distance of a sample from that dataset and then combine the healing process with the SFT. In contrast, for the greatest generality, it's most natural to measure distance and heal with a pretraining dataset that approximates the statistics under which the model was originally pretrained.

Finally, we also investigated an even simpler pruning strategy inspired by analyzing the angular distances across different model families: drop the deepest layers, excluding the final layer before the LLM head, and then (*non-optionally*) heal the damage. For complete clarity, this means that if we are pruning $n$ layers from an $L$-layer model, then we would remove layers $(L - n)$ to $(L - 1)$, inclusive.

# 4 RESULTS

In this section, we demonstrate the effectiveness of our pruning strategy on different question-answering (QA) benchmarks and highlight a robust pruning-driven transition in performance (§4.1), while, in contrast, we find that the autoregressive perplexities of the healed pruned models are continuous across their transition points (§4.2); then, after comparing the similarity statistics between different layers across model sizes and families (§4.3), we contrast our principal similarity-informed pruning strategy with a simpler remove-the-deepest-layers strategy (§4.4).

For our experiments, we pruned a wide variety of large-scale LLMs from 2.7B to 70B parameters spanning 32 to 80 total unpruned layers. Specifically, we used models in the Llama-2 family (Touvron et al., 2023), the Qwen family (Bai et al., 2023), Mistral-7B (Jiang et al., 2023a), and Phi-2 (Javaheripi and Bubeck, 2023). For these models, we executed the "healing" step using QLoRA (Dettmers et al., 2023): our models were quantized to 4-bit precision and then finetuned, using QLoRA for efficient training, on either 164M or 328M tokens from the Colossal Clean Crawled Corpus (C4) (Raffel et al., 2020), a common pretraining dataset. As a result, *each experiment of ours can be performed on a single 40GB A100 GPU*. For our QA evals, we used Massive Multitask Language Understanding (MMLU) (Hendrycks et al., 2020), a common world-knowledge and problem solving benchmark, and BoolQ (Clark et al., 2019), a common yes/no reading comprehension benchmark where the answer has to be inferred from the text itself. The specifics of our models, healing procedure, dataset choices, and evaluation details can be found across Appendix B; ablations of different hyperparameter choices can be found across Appendix C.

## 4.1 ACCURACY ON QA BENCHMARKS

Our first set of results are shown in Figure 2, where we plot 5-shot MMLU accuracy as a function of the fraction of layers removed: in the left panel we present the Llama-2 family, in the middle panel we present models from the Qwen family, and in the right panel we show Mistral-7B and Phi-2. In order to better compare models of different total number of layers, in these plots we opted to normalize the $x$-axis by the fraction of layers removed (rather than the absolute number of layers removed). Note that since MMLU contains multiple choice questions with four possible responses, the expected accuracy of random guessing is 25%.

Importantly, we see a characteristic flat region of robust performance followed by a sharp transition to random accuracy at a pruning fraction around 45%-55% for models in the Llama-2 family, 35% for Mistral 7B, 25% for Phi-2, and 20% for models from the Qwen family. This implies that the essential knowledge required to achieve a model's top score isn't removed by significant layer removal – even though the fraction can be quite large(!) – until eventually that knowledge is lost at a critical

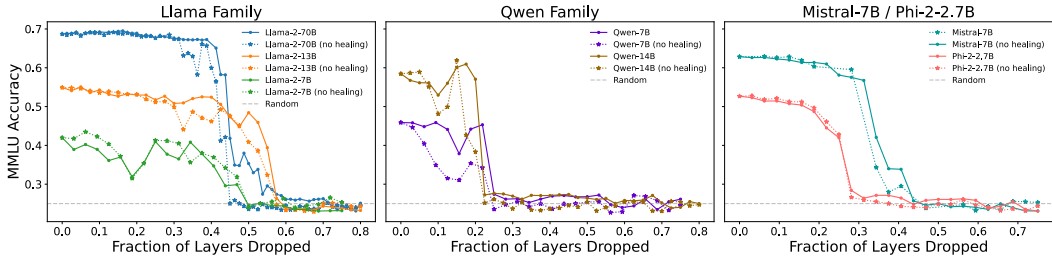

Figure 2: MMLU accuracy (5-shot) vs. fraction of layers dropped for different model families. (*Left:* Llama-2 family; *Middle:* Qwen family; *Right:* Mistral-7B and Phi-2.) The solid lines represent performance after dropping layers and healing, dotted lines show performance after dropping layers only (no healing), and the dashed gray line is the score for guessing randomly. For these models, healing leads to modest improvements, and performances are quite robust until 20%-55% pruning fractions, depending on model family and size, at which point they transitions to random guessing.

model-dependent threshold.[4] Contrasting the curves with and without healing, we see that finetuning offers a modest improvement by better preserving the unpruned performance and pushing the phase transition to random guessing to slightly larger pruning fractions.

Broadly we see that layer pruning is more robust for the larger and deeper models, e.g. Llama-2-13B and Llama-2-70B, which we hypothesize could be related to the fact that either the smaller models are more overtrained, making parameters less redundant, or that the deeper models can afford to lose more layers in an absolute sense. Also, the Qwen family is strange, a fact we will further elaborate on in §4.3.

### 4.2 LOSS ON NEXT-TOKEN PREDICTIONS

In this section, we look at the effect of layer pruning on the pretraining optimization objective – the cross-entropy loss of next-token prediction – when evaluated on a subset of the C4 validation dataset.[5] In order to have a fair comparison across models with different sized vocabularies $V$, we normalize the loss by $\log V$, which corresponds to the loss of sampling tokens randomly with uniform probability. (See Appendix B.2 for more details.)

In Figure 3, we plot the normalized C4 validation loss for all seven of our models, after healing (left panel) and before healing (right panel), as a function of the fraction layers removed. Without healing, we see that there is a somewhat sharp(ish) transition to random guessing for each model at approximately the pruning fraction that the QA benchmark accuracies also sharply transition to random guessing, suggesting that models are hopelessly harmed at this point, cf. Figure 2. Next, contrasting the scales of both plots, we see that healing significantly restores the next-token prediction ability of all the models to near-unpruned levels, with the loss increasing slowly and linearly with layer dropping. Most strikingly – from a scientific perspective – is the post-healing continuity through the pruning fractions where we previously found sharp transitions for the QA benchmarks: this decoupling illustrates one way of disconnecting (or creating a miscalibration) between performance on downstream tasks – such as MMLU and BoolQ – and continuous measures of performance – such as the cross-entropy loss.[6]

---

[4]This effect is rather robust to choice of QA benchmark: in Figure 7 we plot the average 0-shot BoolQ accuracy for our model families and observe analogous behavior.

[5]We make sure that none of the validation data are seen during the healing stage.

[6]This is consistent with Schaeffer et al. (2023) that argued jumps in one kind of metric may not be visible in others.

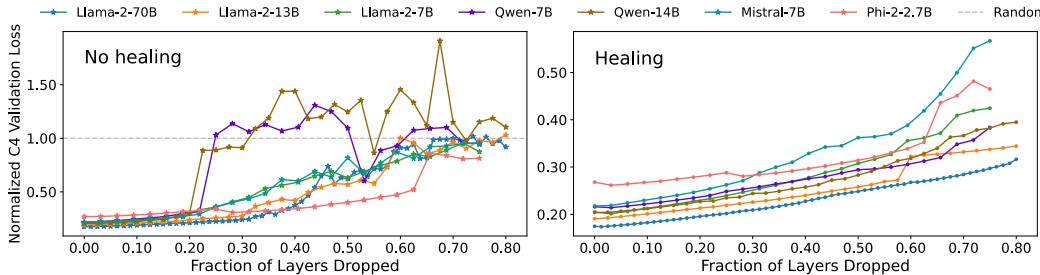

Figure 3: Normalized C4 validation loss vs. fraction of layers dropped before healing (*left*) and after healing (*right*); each curve is normalized by the cross-entropy loss of sampling uniformly from the model's vocabulary. For the experiments before healing, the loss for each model transitions to random guessing (gray dashed line) at approximately the same pruning fractions that the QA benchmarks transition to random guessing; after healing, there is continuity through the regions of sharp transition on QA tasks, cf. Figure 2. Contrasting the overall scale of both plots, it's clear that healing significantly restores the performance on next-token prediction to near-unpruned levels.

### 4.3 ANGULAR DISTANCES BETWEEN REPRESENTATIONS

Given the central role the angular distance (7) plays in our pruning strategy, let's take a subsection to look at these distances across our seven models. For this analysis, the angular distances for each model were averaged over 10k samples from the C4 validation set.

Recall from earlier Figure 1(c): for Llama-2-70B this plotted the angular distance $d(x^{(\ell)}, x^{(\ell+n)})$ that compared the $\ell$-th layer to the $(\ell + n)$-th layer, across all initial indexes $\ell$ for block sizes from $n = 1$ to $n = 64$; the minimum of the curves, $\ell^\star(n)$, gave the optimal block to prune for a given $n$, cf. (6).

A more compact way to display this same data is shown in the heat maps of Figure 4: each square is colored to depict the row-normalized angular distance between layer $\ell$ and $\ell + n$ across all possible $\ell$, and $n$ up to very large fractions of the total number of layers; the optimal layer to prune for a given block size, $\ell^*(n)$, corresponds to the minimal distance in each row.

Across models, we make two generalizations: *(i)* the smallest distances are found across the deeper blocks, meaning deeper layers are typically quite similar to each other and can be more easily dropped; *(ii)* the distances across the deepest blocks – the blocks that include the last layer – take either maximal or nearly-maximal values, meaning one should never drop the final layer. While broadly true, there are a few exceptions. For some models, e.g. Phi-2-2.7B, or for the largest blocks in some models, e.g. Llama-2-7B, final *few* layers seem important. As previously noted, the Qwen family is somewhat unusual: here we see that there are a few odd "islands" of high similarity for shallow blocks; this likely explains the shorter region of robust performance in Figure 2.

### 4.4 A SIMPLER PRUNING STRATEGY

Inspired by our recent conclusions, we experiment with a very simple heuristic pruning strategy: *(1)* if pruning $n$ layers from an $L$-layer model, drop layers $(L - n)$ to $(L - 1)$ so as to remove the deepest block that excludes the final layer; then *(2)* heal with a small amount of finetuning as before. Compared with our principal similarity-informed pruning strategy, this simpler heuristic algorithm has the advantage of never requiring practitioners to load onto a GPU or inference the unpruned model. It also provides a meaningful ablation of the importance of optimizing the block to prune.

In Figure 5, we contrast our two pruning strategies, both before healing (left panels) and after healing (right panels), for the QA benchmarks (MMLU/BoolQ, top/middle panels) and the autoregressive loss (C4 validation, bottom panels). On the one hand, the simple heuristic performs quite poorly without healing the damage incurred by pruning: accuracy on the QA benchmarks decays rapidly to (near-) random with increased pruning fraction, and the loss begins to increase very rapidly even with small amounts of pruning. On the other hand, the results for the two pruning strategies across evaluations

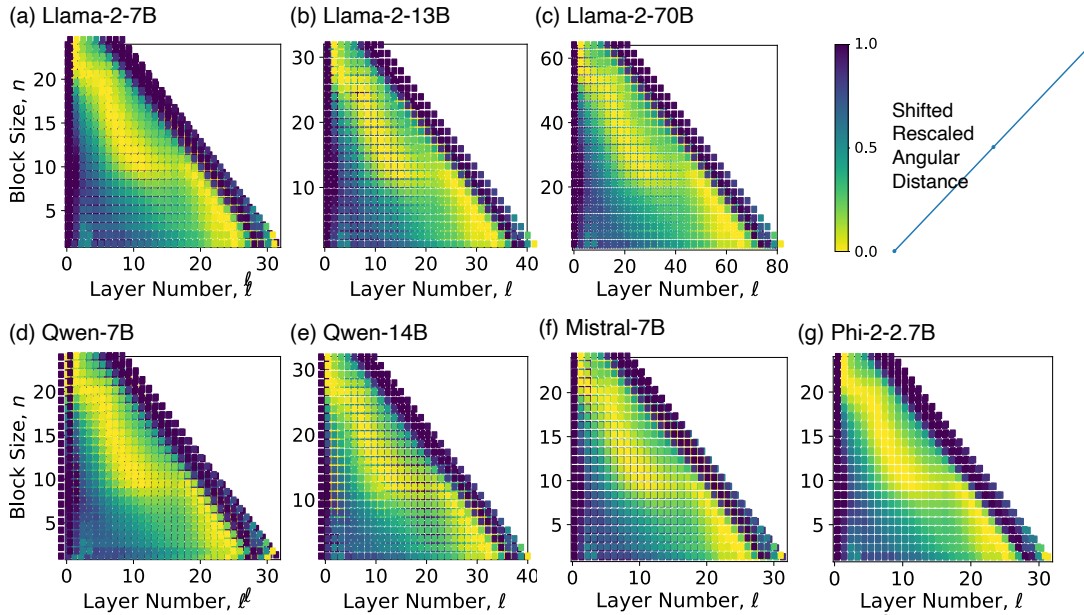

Figure 4: Normalized angular distance (7) from initial layer $\ell$ (x-axis) with block size $n$ (y-axis) for each of the seven models we evaluated; the distance for each $n$ is shifted and rescaled to span the same range, $[0, 1]$ (yellow to purple): the optimal block to prune, $\ell^*(n)$, corresponds to the deepest yellow for each row. Across models, the deeper layers tend to be very similar, though the deepest blocks that include the final layer (squares along the outer diagonal) are (near-)maximally dissimilar.

are quite comparable after healing: for the QA benchmarks, the similarity-informed algorithm slightly better preserves the accuracy before the phase transition, though the simple algorithm perhaps pushes the phase transition to slightly greater pruning factions; and for the loss, the curves nearly lie on top of each other, though the similarity-informed strategy does marginally outperform for all amounts of pruning. These experiments are strong evidence that the purpose of post-pruning finetuning is the healing of damage at the pruning interface and not the acquisition of additional knowledge.

## 5 DISCUSSION AND FUTURE DIRECTIONS

At the end of this work, many readers are puzzled by the following: are the deeper layers entirely useless? So far, we've provided evidence that the elimination of the deeper layers does not affect performance on QA tasks like MMLU (Figure 2), while at the same time have shown that their removal does disrupt the next-token predictions of the underlying model (Figure 3). Since perplexity often correlates with performance on downstream tasks, which are the tasks that are hurt by layer pruning?

Here are two hypotheses consistent with the fact that the model's perplexity is disturbed proportionally to pruning fraction:

*(i)* The deeper layers are not essential for storing knowledge, but are useful for more complicated computations, such as those that involve reasoning.

*(ii)* The deeper layers are necessary when the model has to generate many tokens before answering a question, such as when it produces a chain-of-thought (CoT).

We test these hypotheses by evaluating our layer-pruned models on tasks that involve CoTs or reasoning. For the former, we'll look at Chain-of-Thought MMLU (CoT-MMLU); for the latter, we'll

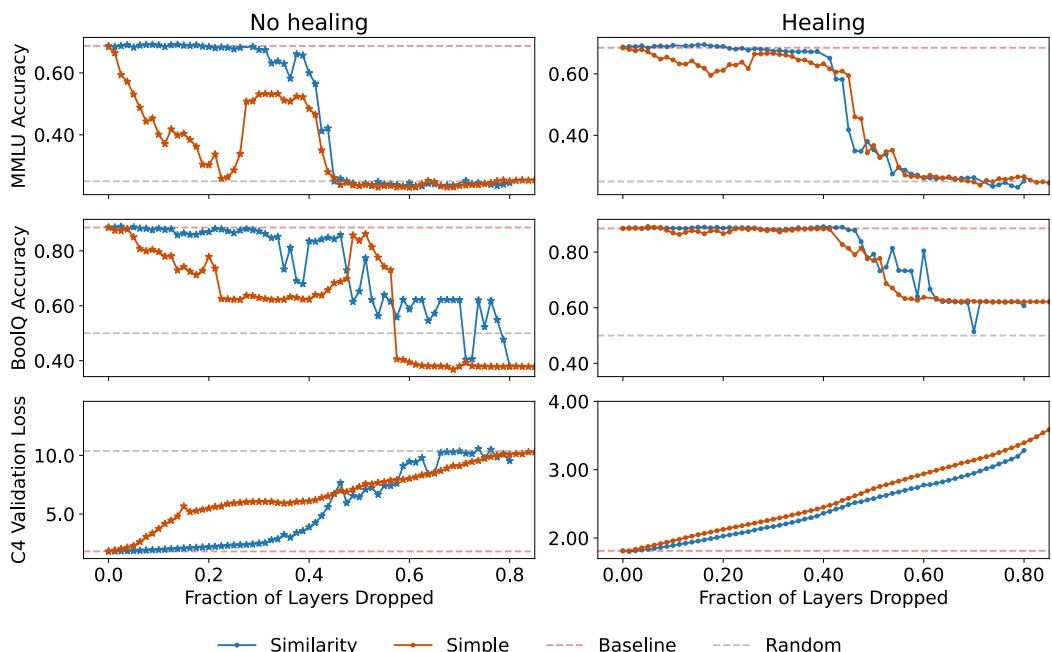

Figure 5: Evaluation of Llama-2-70B with the simple pruning heuristic (solid red line), shown along with scores for the similarity-informed pruning strategy (solid blue line), scores of the unpruned Llama-2-70B (red dashed line), and scores for randomly guessing (gray dashed line). (*Left:* before healing, *Right:* after healing; *Top:* MMLU, *Middle:* BoolQ, *Bottom:* C4 Validation Loss.) Without healing, the simple heuristic performs poorly across all evals; with healing, the scores of both methods are quite similar.

look at GSM8K (Cobbe et al., 2021), a grade-school math benchmark, and HellaSwag (Zellers et al., 2019), a multiple choice common-sense reasoning benchmark.[7]

In Figure 6, we plot the performance of Llama-2 70B pruned with the similarity-informed pruning strategy across CoT-MMLU (left), GSM8K (center), and HellaSwag (right): on the one hand, both GSM8K and HellaSwag, our two reasoning tasks, exhibit immediate degradation in performance with any amount of pruning, correlating with a similar decrease in the perplexity evals (Figure 3); on the other hand, CoT-MMLU shows a relatively flat region of robust performance with pruning, analogous to our previous results on QA benchmarks (e.g. Figure 2). This is some initial evidence for hypothesis *(i)* over hypothesis *(ii)*: the deeper layers may be useful for higher-level reasoning tasks, while less important for knowledge intensive QA tasks; moreover, perplexity errors due to pruning do not compound to hurt QA evals when the model is required to generate many tokens.

Now at the conclusion of the work, we are left with the following questions:

---

[7]Here are the details for how we performed these three evaluations:

- For **CoT-MMLU**, we followed the `flan_cot_fewshot` evaluation in EleutherAI (Gao et al., 2023), in which models produce a chain of thought before generating their answer. Note that the accuracy at 0% pruning fraction for MMLU without CoT is much better than the analogous accuracy at 0% pruning fraction for CoT-MMLU ($\sim 69\%$ vs. $\sim 43\%$, respectively; cf. Figures 2 and 6), consistent with some previous work (e.g., see Table 16 of Chung et al. (2024)).

- For **GSM8K**, we used the `gsm8k_cot` evaluation in EleutherAI (Gao et al., 2023) and measured `pass@1`; for each problem we extracted an answer from a single generation (with CoT) and checked for correctness against the ground-truth answer.

- For **HellaSwag**, we used the `hellaswag` evaluation in EleutherAI (Gao et al., 2023). Note that HellaSwag is a multiple-choice benchmark, so random performance is 25%.

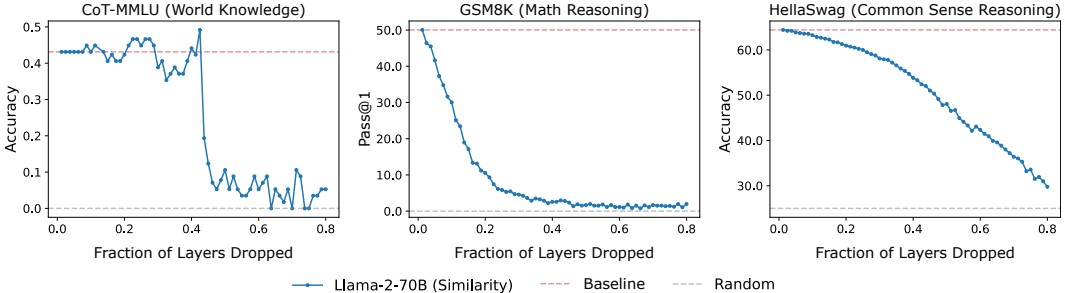

Figure 6: Evaluation of Llama-2 70B with the similarity-informed pruning strategy across different evaluation tasks. (*Left:* Chain-of-Thought MMLU (CoT-MMLU), *Center:* GSM8K, *Right:* HellaSwag.) We see that GSM8K and HellaSwag show immediate degradation of performance with any level of pruning, while CoT-MMLU behaves qualitatively similarly to MMLU without CoT; this suggests that the deeper layers are likely necessary for reasoning tasks.

- What are better layer-pruning strategies? What are better approaches to healing?[8]
- Why does healing eliminate the phase transition in the loss but not in the QA accuracies?
- With more comprehensive evals, will accuracy on different tasks degrade at different depths?
- Relatedly, is knowledge generally stored in shallow or middle layers, or is it delocalized?
- Can we devise a pruning strategy that is robust for reasoning tasks?
- Do pretraining details affect the ability to prune, e.g., are scaling-law over-trained or distilled models more difficult to prune?
- How can we enable LLMs to more effectively use the parameters in their deepest layers?

Some of these questions would benefit from studying both layer similarity and pruning across different pretraining checkpoints; for instance, at what point does the sharp phase transition and critical depth in the QA accuracies emerge, and does more training lead to better use of the prunable parameters? Others suggest explorations with different pretraining architectures and objectives, e.g. in order better make use of the deeper layers (for example, one can imagine applying layer dropout (Fan et al., 2019) or early exit during pre-training (Elhoushi et al., 2024) to induce equal usage of layers). With more comprehensive evaluations, if different kinds of QA tasks degrade at very different depths, then this might indicate that the knowledge required to complete those tasks is stored across different layers.[9] It would be very interesting to use pruning to systematically study these kind of interpretability questions.

## ACKNOWLEDGMENTS AND DISCLOSURE OF FUNDING

We thank Aaron Schwartz for his initial collaboration, Aaditya Singh and Sho Yaida for discussions, and Aaditya Singh for comments on the draft. We would also like to acknowledge the 2023 NeurIPS Large Language Model Efficiency Challenge for initializing us for work on this project. A.G. is supported by the NSF CAREER grant DMR-2045181, the Sloan Foundation, and by the Laboratory for Physical Sciences through the Condensed Matter Theory Center. D.R. acknowledges support from

---

[8]At the cost of introducing another hyperparameter and requiring both pruned and unpruned models to fit in memory during finetuning, one natural way to improve healing is by adding an auxiliary student-teacher loss that explicitly addresses the pruning mismatch (5), such as

$$\mathcal{L}_{\text{aux}} \sim \left( x^{(\ell^*+n)}(\theta_0) - x^{(\ell^*)}(\theta) \right)^2 , \tag{8}$$

where $\theta_0$ are the frozen parameters of the unpruned model, and $\theta$ are the parameters of the pruned model to be healed; thus, $x^{(\ell^*+n)}(\theta_0)$ is the input to the $(\ell^*+n)$-th layer in the unpruned model, $x^{(\ell^*)}(\theta)$ is the input to that same layer after pruning, and $\mathcal{L}_{\text{aux}}$ minimizes their mismatch. We thank Sho Yaida for this observation.

[9]Alternatively, one could measure $d(x^{(\ell)}, x^{(\ell+n)})$ or find $\ell^*(n)$ as a function of different eval datasets.

the National Science Foundation under Cooperative Agreement PHY-2019786 (the NSF AI Institute for Artificial Intelligence and Fundamental Interactions, http://iaifi.org/) and appreciates both the sanction and support of Sequoia Capital. This paper has been brought to you residually by the letters $G$, $P$, and $U$, after summing over many layers.

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
