# A  EXTENDED LITERATURE REVIEW

In this section, we review practical strategies for post-training efficiency and discuss some scientific investigations that provide motivation for, or insight into, our approach: in §A.1, we first review the history of pruning and then discuss its modern application to LLMs; in §A.2, we contrast pruning with distillation, an alternative strategy for reducing the parameter count of LLMs; then in §A.3, we discuss the various practical methods for efficient finetuning and inference acceleration that can be used in conjunction with our pruning strategy; finally in §A.4 we highlight some scientific investigations into some depth-dependent statistical properties of LLMs that are complementary to our results.

## A.1  PRUNING

*Pruning* is a method for reducing the size of a trained machine-learning model by removing unnecessary parameters, either individually or together as a group. Pruning for neural networks has a long history (LeCun et al., 1989; Hassibi and Stork, 1992), and, as originally conceived, *unstructured pruning* techniques sparsify networks by removing individual parameters based on pre-defined criteria. For instance, if a parameter of the model has a very small value, then removing it – i.e. by setting it to exactly zero – will likely have minimal impact on performance. Inspired by this early work, modern researchers began exploring different criteria for such unstructured pruning, focusing mostly on computer vision models (Han et al., 2015; Chen et al., 2015; Srinivas and Babu, 2015). In particular, Han et al. (2015) developed an *iterative pruning* method for alternatively pruning and finetuning a network in order to reach better compression ratios and performance.

While these models were smaller, they were not necessarily more efficient: sparsifying networks by removing individual parameters according to a criterion leads to irregular or pseudorandom sparsification patterns that are difficult to accelerate without specialized hardware or libraries designed for sparsity (Li et al., 2016). To that end, *structured pruning* techniques were developed to remove irrelevant groups of parameters together, such as particular channels or filters in convolutional networks. As this increased their practical relevance, researchers then began exploring structured pruning across computer vision (Li et al., 2016; Wen et al., 2016; Hu et al., 2016; He et al., 2017; Huang et al., 2018) and pre-transformer NLP architectures (Murray and Chiang, 2015; See et al., 2016; Kim and Rush, 2016).

Following unprecedented progress in language modeling, recent work has focused on applying structured pruning methods to the Transformer (Vaswani et al., 2017). These studies consider nearly every possible component of the model architecture for elimination, with methods ranging from dropping attention heads (Voita et al., 2019; Michel et al., 2019; Kim and Awadalla, 2020), to dropping layers (Fan et al., 2019; Zhang and He, 2020; Fan et al., 2021; Jha et al., 2023; Sajjad et al., 2023; Liu et al., 2023a), to pruning hidden states (Hou et al., 2020), to rank reducing large weight matrices (Sharma et al., 2023), replacing sparse weight matrices with smaller dense ones (Ashkboos et al., 2024), to many combinations of the aforementioned groups (Xia et al., 2022; Lagunas et al., 2021).

Of the prior work that also considers transformer layer dropping, most (Fan et al., 2019; Zhang and He, 2020; Fan et al., 2021; Sajjad et al., 2023; Xia et al., 2022) study BERT-style models (Devlin et al., 2018), while we consider decoder-only GPT-style models (Radford et al., 2019) that are most commonly used for large-scale language modeling and generation. BERT-style models are naturally suited for understanding tasks due to their bidirectional masked language modeling (MLM) objective, while GPT-style models are instead suited for generation, due to their autoregressive objective. While this divide has been questioned in light of more powerful GPT-style models (Zhong et al., 2023), previous work (Ethayarajh, 2019) has found significant qualitative differences between BERT and GPT models in terms of the evolution of the layer-wise representation of words. Altogether, this suggests that layer-dropping strategies will behave differently between the two families.

One study for BERT-style pre-trained models, Sajjad et al. (2023), concludes that the best layer-pruning strategy is dropping the final layers; this partially resonates with our results, although in contrast we find that *(a)* for some pruning sizes keeping the last few layers of the model is actually beneficial, and that *(b)* for all pruning sizes keeping the very last layer is essential. Additionally, while the authors also study similarity between representations in different layers – as in our approach

– they actually found a higher similarity between representations in the shallow layers compared to the deeper ones – which very sharply disagrees with our results. Importantly, the models considered in Sajjad et al. (2023) consist of a few hundred million parameters, which is much smaller than the model scales we consider in our work. Perhaps as a consequence, the authors didn't observe the sharp transition in downstream accuracies that we report in §4.1, despite the fact that they also finetuned their pruned models.

In contrast, while Jha et al. (2023) does consider GPT-style models, the methodology is quite different from ours: *(i)* rather than pretraining first and then using a fixed layer-dropping strategy as we do, instead the authors incrementally drop layers in a modified pretraining procedure; and *(ii)* the authors study their own sub-1B parameter models, while we focus on the families of readily available, open-weight, large-scale 2.7B-70B parameter models that are commonly used and/or finetuned for practical applications.

As we were finalizing our preprint, Men et al. (2024) was posted: this paper empirically studies different layer-pruning strategies for GPT-style models (Llama-2 7B and Baichuan2-7B-base) and their subsequent effects on benchmarks (MMLU, CMMLU, and CMNLI). They investigate various layer-importance metrics – notably, their "Block Influence" function is similar to our cosine similarity metric – and find that they are able to prune up to ∼28% of layers of Llama-2 7B with minimal impact on performance. This provides independent evidence supporting our main takeaway that the deeper layers are not critical for storing knowledge.

Finally, a systematic approach to layer dropping in transformers has also been studied in the context of *wav2vec* models, which are encoder-only models that map speech to embeddings and are sized in the hundred-million parameter regime (Baevski et al., 2020). With these models, Liu et al. (2023a) developed a layer-pruning algorithm based on the correlation between layers and downstream metrics. Beyond the model architecture and domain, one significant difference between this and our work is that Liu et al. (2023a) considered non-contiguous pruning proposals, e.g. dropping alternate layers. Our intuition for layer pruning predicts that this shouldn't work as well – at least for decoder-only language models – as it creates multiple mismatches, one with each block of layers removed.

## A.2    MODEL DISTILLATION

A completely different method for reducing the size of a trained machine-learning model is *model distillation* (Hinton et al., 2015), in which knowledge is transferred from a large "teacher" model to a smaller "student" model by training the student on the distribution predicted by the teacher. The essential insight is that this can transform the very general knowledge and capabilities of the teacher into more streamlined, compressed, and possibly skill-specific representations.

While a very general technique, in the setting of language models, distillation has been implemented with *(a)* white-box approaches, in which the the student is trained to imitate the teacher's logits (Gu et al., 2023) or hidden states (Jiao et al., 2019); as well as with *(b)* black-box approaches, in which the student only has access to the output tokens generated by the teacher. This latter approach broadly covers cases where the student is trained on text that is augmented by the teacher in some way, such as by adding synthetic labels (Wang et al., 2021), generating high quality synthetic text (Eldan and Li, 2023; Li et al., 2023a, Gunasekar et al., 2023) by providing chain of thought reasoning (Fu et al., 2023; Hsieh et al., 2023), which aims to enhance the student's reasoning skills, or by annotating instructions that enhance the student's instruction-following capabilities (Jiang et al., 2023b).

Compared to layer pruning, these distillation methods require considerable computational resources due to the reliance on the large teacher to process a big corpus of data. Instead, our similarity-based pruning strategy only requires computing the similarity between representations at different layers on a small subset of a pretraining corpus, while our second simpler pruning strategy only uses the reduced model post pruning.

## A.3    EFFICIENT FINETUNING AND INFERENCE ACCELERATION

Complementary to directly reducing size of a model, *parameter-efficient finetuning* (PEFT) focuses on reducing the cost of specializing LLMs to certain tasks. In particular, Low Rank Adapters (LoRA) reduce the memory and compute of fine tuning by freezing the pretrained model and introducing a parametrically small number of additional trainable weights (Hu et al., 2021). We use its quantized

cousin, QLoRA (Dettmers et al., 2023), to keep our experiments cost efficient. Other PEFT methods that can be combined with our work are Li et al. (2023b) and Zhang et al. (2023): in the first, the initialization of the LoRA matrices is adjusted to a quantization scheme; in the second, LoRA ranks for different LLM modules are chosen in an adaptive manner.

For additional efficiency gains we could combine our layer-pruned models with methods that further accelerate inference: with speculative decoding (Leviathan et al., 2023), tokens are rapidly generated from a smaller draft model and then evaluated in parallel by the main model; with Medusa (Cai et al., 2024) the draft model is discarded for extra decoding heads, but ultimately achieves a similar effect. In particular, it could be interesting to consider highly-compressed layer-pruned models as potential draft models in a speculative decoding setup.

### A.4 A BREADTH OF DEPTH-DEPENDENT STUDIES

Finally, let us highlight some scientific work that study the depth-dependent properties of LLMs. One relevant direction considers how knowledge and linguistic properties are encoded in language models. On the one hand, Meng et al. (2022) and Dai et al. (2021) analyze the *storage and recall* of factual associations: these works emphasize that knowledge localizes within the middle (Meng et al., 2022) or final (Dai et al., 2021) layers, which has implications for directly editing or erasing part of a model's factual knowledge. On the other hand, attempts to perform such editing gives evidence that information may be stored non-locally across layers (Hase et al., 2023). Relatedly, Geva et al. (2023) investigates the way facts are *processed* during inference, distinguishing between the role of attention heads, for attribute extraction, and the MLP blocks, for subject enrichment: both are delocalized across several layers.

Next, following the earlier "logic lens" (nostalgebraist, 2020), Belrose et al. (2023) invented a technique they called "tuned lens" to study the *trajectory of predictions* by using a learnable affine transformation to convert intermediate representations into a distributions over tokens (see also Din et al. (2023)). By studying the layer-to-layer dynamics of this distribution, the authors noted that it tended to converge. This convergence is very suggestive that that the deeper layers could be prunable, while the fact that they had to train an affine probe is likely related to our observation that the final layer cannot be pruned. Somewhat relatedly, Gurnee and Tegmark (2023) observed that geographic features in the underlying text can be determined from linear probes trained on intermediate activations, as long as the activations are deeper than halfway.

More abstractly, Voita et al. (2023) and Liu et al. (2023b) found that the sparsity of activations transitions at around halfway through a network's forward pass, evolving from sparse to dense. Perhaps relatedly, Panigrahi et al. (2023) investigated which model weights update the most during finetuning, finding that it's those in the mid-layers.

Altogether, these deep studies are complementary to our work, which, on the one hand, provides evidence that removing the deepest layers of an LLM does not significantly alter the model's performance, and, on the other hand, demonstrates a sharp pruning transition after removing approximately half of an LLM's deepest layers.

## B EXPERIMENTAL DETAILS

Here we explain various details of models and healing (§B.1) and of evaluations (§B.2).

### B.1 MODEL AND HEALING DETAILS

All models in this paper were fine-tuned using the Hugging Face `Trainer` API (Wolf et al., 2020). A list of models and their paths on Hugging Face are as follows:

| Model | ‖ | Repository Path |
|---|---|---|
| Llama-2 7B | ‖ | `meta-llama/Llama-2-7b-hf` |
| Llama-2 13B | ‖ | `meta-llama/Llama-2-13b-hf` |
| Llama-2 70B | ‖ | `meta-llama/Llama-2-70b-hf` |
| Mistral 7B | ‖ | `mistralai/Mistral-7B-v0.1` |
| Phi-2 (2.7B) | ‖ | `microsoft/phi-2` |
| Qwen 7B | ‖ | `Qwen/Qwen-7B` |
| Qwen 14B | ‖ | `Qwen/Qwen-14B` |

For healing, we used the version of the Colossal Clean Crawled Corpus (C4) (Raffel et al., 2019) from Hugging Face: `data = load_dataset("c4", 'en')`. We truncated long examples as described later in the paragraph and added special tokens when available.[10] Models were finetuned for 5000 steps with a global batch size of 16: this corresponds to total finetuning tokens of $16 \times 5000 \times$ [`max_seq_length`] for each model. We used a cosine-annealed learning rate schedule, with a warmup of 100 steps. When possible, the peak learning rate was set to the peak learning rate from the model's pretraining; in practice, this means all models were trained with a peak LR of 3e-4, with the exceptions of Phi-2 (Javaheripi and Bubeck, 2023), which was trained with a peak LR of 2e-4 during pre-training, Llama-2-70B, which was trained with a peak LR of 3e-5 (a value that resulted from a sweep), and Mistral-7B which was trained with a peak LR of 3e-6 (also a value that resulted from a sweep). All models 7B parameters or smaller were trained with a max sequence length of 2048 tokens, while all models 13B parameters or greater were trained with a max sequence length of 4096 tokens. While we realize that some models may have been pretrained on longer sequences, e.g. Qwen-*the-outlier* (Bai et al., 2023), we decided to the max sequence length consistent across models of similar size to allow fairer comparisons across model families.

On top of the Hugging Face Trainer API, we used quantization and Low-Rank Adapters (LoRA) (Hu et al., 2021) for all of our finetuning:

- For quantization, we used the `bitsandbytes` library for QLoRA (Dettmers et al., 2023) to quantize our models to 4 bits.

- For LoRA, we used the Hugging Face `peft` library (Mangrulkar et al., 2022). We set the LoRA dropout to 0.05 and kept the LoRA $\alpha$ equivalent to the LoRA rank, following (Lee et al., 2023). Aside from two exceptions, discussed below, models are trained with LoRA rank 64.

- Also following Lee et al. (2023), we only applied LoRA to FFN modules: `["gate_proj", "down_proj", "up_proj"]` for Llama-2 and Mistral models, `["fc1", "fc2"]` for Phi-2, and `["w1", "w2", "c_proj"]` for Qwen models.

The large majority of these hyperparameter choices are standard and found in previous works, e.g. Lee et al. (2023) and Dettmers et al. (2022). For absolute clarity, we list display all the model specific architecture and healing details below:

| Model | # Layers | Vocab Size | Max Seq. Len. | FT Tokens | Peak LR | LoRA Rank |
|---|---|---|---|---|---|---|
| Llama-2 7B | 32 | 32,000 | 2048 | 164M | 3e-4 | 2 |
| Llama-2 13B | 40 | 32,000 | 4096 | 328M | 3e-4 | 64 |
| Llama-2 70B | 80 | 32,000 | 4096 | 328M | 3e-5 | 8 |
| Qwen 7B | 32 | 151,936 | 2048 | 164M | 3e-4 | 64 |
| Qwen 14B | 40 | 151,936 | 4096 | 328M | 3e-4 | 64 |
| Mistral 7B | 32 | 32,000 | 2048 | 164M | 3e-6 | 4 |
| Phi-2 2.7B | 32 | 51,200 | 2048 | 164M | 2e-4 | 64 |

We also have the following hyperparameters common between all models:

---

[10]N.B. the Qwen tokenizer from Hugging Face does not include any special tokens; in this case, it was essential to add a default padding token.

| Config | Value |
|---|---|
| Finetuning dataset | C4 |
| Batch size | 16 |
| LoRA $\alpha$ | LoRA rank |
| LoRA dropout | 0.05 |
| LoRA targets | FFN modules |
| LR scheduler | Cosine |
| Warmup steps | 100 |
| Total steps | 5000 |

## B.2 EVALUATION DETAILS

We performed three principal evaluations: accuracy on *MMLU*, accuracy on *BoolQ*, and loss on *C4*.

For **MMLU accuracy**:

- We use the `cais/mmlu` version of the dataset from Hugging Face.

- We follow the formatting suggested in the original reference (Hendrycks et al., 2020) without further prompt engineering.

- For constructing few-shot examples, we use the `dev` set from `cais/mmlu`.

- For our experiments, we use 0 few-shot examples; our results and analysis are robust to this choice, cf. Figure 8.

- We report average accuracy across all subjects.

For **BoolQ accuracy**:

- We used the `hassansh/boolq_n_shot` version from Hugging Face.

- For our experiments, we use 0 few-shot examples.

- The complete BoolQ results – truncated from the main text – are shown here in Figure 7: in the left panel we present the Llama-2 family, in the middle panel we present models from the Qwen family, and in the right panel we should Mistral-7B and Phi-2; we also make the experiments without healing semi-transparent in order to better display the results from the complete similarity-informed pruning method. Importantly, while we see here that healing plays a more important role than it did for MMLU in Figure 2, after healing we still have a characteristic flat region of robust performance; as before, the capabilities required to achieve a model's top score isn't removed by significant layer pruning until a critical model-dependent threshold.

For **C4 Validation Loss**:

- We used the `c4` version from Hugging Face (soon be deprecated in favor of `allenai/c4`).

- We evaluated using the *validation* split as we healed with the train split.

- Given its size, we randomly sampled 60k sequences and held them fixed across all models.

- In Figure 3 we normalized the loss to facilitate fair comparison across model families that employ different vocab sizes: to normalize, we divided by $\log V$, where $V$ is the *per-model* vocab size (listed in a table in §B.1). This, $\log V$, corresponds to the loss of sampling tokens uniformly, which naturally sets the scale for a given model.

## C ABLATIONS

Here we detail various ablations: prompting (§C.1), finetuning seed (§C.2), LoRA rank (§C.3), other pruning strategies (§C.4). Qualitatively, the results of the paper are quite robust to the variation of any of these.

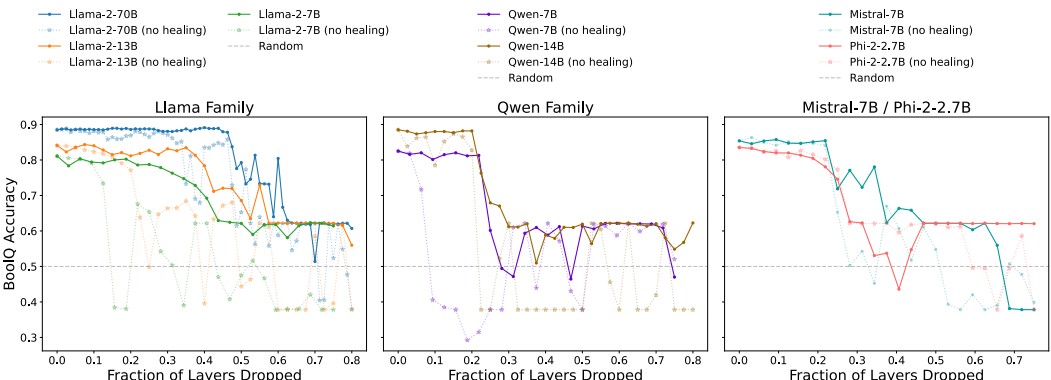

Figure 7: BoolQ accuracy (0-shot) vs. fraction of layers dropped for different model families. (*Left:* Llama-2 family; *Middle:* Qwen family; *Right:* Mistral-7B and Phi-2.) The solid lines represent performance after dropping layers and healing, and the (semi-transparent) dotted lines show performance after dropping layers only (no healing), and the dashed gray line is the score for guessing randomly. For BoolQ, healing leads to important improvements such that performances; then, across all models, performances are quite robust until 20%-55% pruning fractions, depending on model family and size, at which point they transitions to random guessing.

## C.1 PROMPTING

It's common knowledge that altering the prompt on QA evaluations can significantly impact results. To control for prompting, we ablate the MMLU accuracy for our principal similarity-informed pruning described in §3.2 when applied to Llama-2-13B: in the left panel of Figure 8, we show results for changing the ordering of the few-shot examples in the prompt, and in the right panel the same figure, we show results for changing the number of few-shot examples. Broadly we see that the layer-pruning method is robust to these changes.

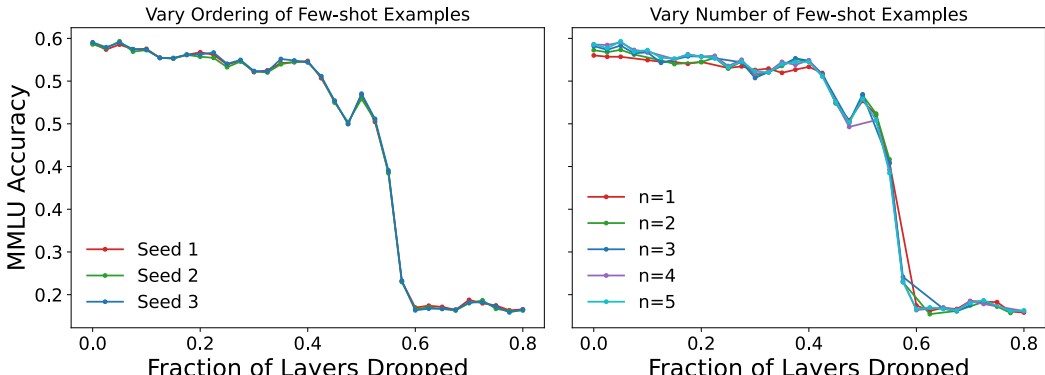

Figure 8: Effect of prompt ablations on MMLU accuracy vs. fraction of layers dropped for Llama-2-13B. *Left:* We vary the ordering of the few-shot examples and see it does not have any impact. *Right:* We very the number $n$ of few-shot examples; while careful study of the flat region suggests increasing the number of few-shot examples marginally improves performance, regardless, the layer-pruning strategy is robust to this kind of variation.

## C.2 FINETUNING SEED

Here we vary the finetuning seed. For all of our experiments, we use the following code snippet to ensure reproducibility:

```
SEED_VAL = 0
transformers.enable_full_determinism(SEED_VAL)
```

Since we begin with a pretrained model, the finetuning seed doesn't affect initialization, but it will impact the stochastic aspects of further training such as data order. To control for this, we ablate the finetuning seed for our principal similarity-informed pruning described in §3.2 when applied to Llama-2-13B: in Figure 9 we observe that the layer-pruning method is robust to the choice of seed.

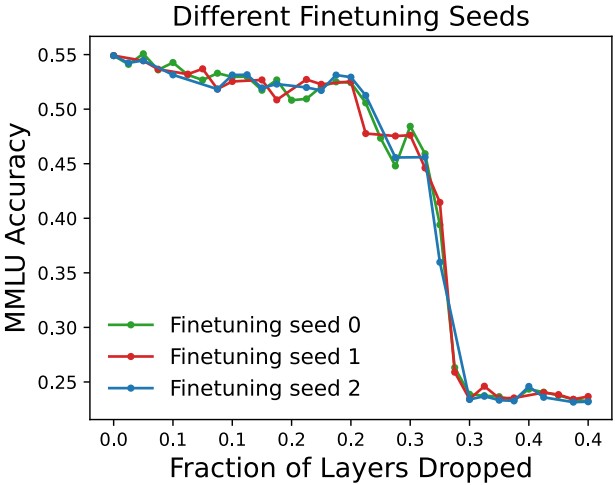

Figure 9: Effect of varying the finetuning seed on MMLU accuracy vs. fraction of layers dropped for Llama-2-13B: there is no meaningful effect.

## C.3 LoRA RANK

Here we vary the LoRA rank used for healing. Unfortunately, our compute budget did not allow us to make an exhaustive sweep across all of our experimental configurations. In lieu of that, we employed the following protocol for our main experiments:

- Begin with rank 64, following the QLoRA setup (see, e.g. Appendix B.2 of Dettmers et al. (2023)).
- If healing with that rank significantly harms the performance compared to no healing, then sweep LoRA ranks for that model and, for the other evaluations, pick the best performing LoRA rank according to its MMLU accuracy.

This protocol is designed to maximize the chance that healing will improve performance across all of our evaluations. For simplicity, we ran this rank-picking protocol using the simple pruning heuristic, with the exception of Llama-2-70B.

In practice, this led to us using rank 64 for every model with the exceptions of Mistral-7B, with rank 4, Llama-2-7B, with rank 2, and Llama-2-70B, with rank 8. (To review this same information in tabular form, see the second Table in §B.1.) Figure 10 displays the sweeps over MMLU accuracy supporting these choices for Mistral-7B (bottom left panel), Llama-2-7B (bottom middle panel), and Llama-2-70B (top right panel): overall, while the LoRA rank does not have a significant impact on the qualitative behavior of the healed model, decreasing the LoRA rank generally improves performance. In the top left and middle panels of Figure 10, we show corresponding sweeps for

Mistral-7B (top) and Llama-2-7B (middle) using the similarity-informed pruning strategy: we see that for this pruning method both models are much more robust, though rank 2 is still the top performing rank for Llama-2-7B.

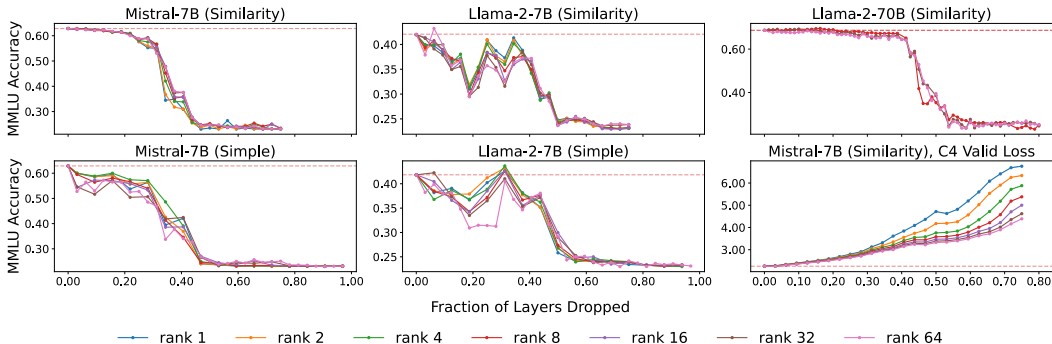

Figure 10: Effect of varying the LoRA rank. **Top**: 5-shot MMLU accuracy vs. fraction of layers dropped using the similarity-informed pruning strategy on Mistral-7B (*left*), Llama-2-7B (*middle*), and Llama-2-70B (*right*). Across all ranks we observe similar behavior, though there's a small effect of decreasing rank improving overall performance. **Bottom, left and middle**: 5-shot MMLU accuracy vs. fraction of layers dropped using the simple pruning heuristic on Mistral-7B (*left*) and Llama-2-7B (*middle*). As before, qualitative behavior is similar across ranks, though in this case it's much clearer that decreasing rank improves performance. **Bottom, right**: C4 validation loss vs. fraction of layers dropped using the similarity-informed pruning strategy on Mistral-7B. In contrast to MMLU, decreasing rank harms performance; together, these results suggest that larger ranks may be overfitting.

The characteristic improvement of MMLU accuracy with decreasing LoRA rank – even for extremely low ranks(!) – deserves an explanation. One possibility is that lowering the LoRA rank can better regularize finetuning against overfitting. In particular, astute readers may have been surprised at the discussion of peak learning rates in §B.1: models were finetuned with the same peak used in pretraining; a "large" LoRA rank of 64 introduces a number of additional parameters that may overfit to C4. This overfitting would certainly be harmful, since the actual pretraining datasets for the models we consider are *(a)* unknown to us, and *(b)*, likely to be of significantly higher quality than C4.

We investigate this directly for Mistral-7B. In the bottom right panel of Figure 10 we plot the C4 validation loss across different LoRA ranks: we see that while decreasing the LoRA rank generally improves MMLU accuracy (cf. left-most panels), at the same time it harms the C4 validation loss. This supports our overfitting hypothesis. In a greater-resourced future, it would be interesting to improve the healing process by considering other forms of regularization and learning rate tuning.

### C.4 OTHER PRUNING STRATEGIES

Here we study how the similarity-informed pruning strategy (§ 3.2) compares to other layer-pruning baselines: specifically, we contrast with pruning random layers and pruning shallow layers. In Figure 11, we observe that the similarity-informed strategy from the main text outperforms both of these other strategies on an MMLU evaluation of Llama-7B.

## D BROADER IMPACTS

This work studies the pruning of open-weight LLMs. Positive societal impacts include an increased understanding of how LLMs process information across layers as well as the demonstration of potential practically useful techniques for improving the efficiency of LLM inference. Negative societal impacts are minimal; however, there may be possible second-order negative effects given that LLM systems are tools that can be used both positively and negatively, given different downstream use cases.

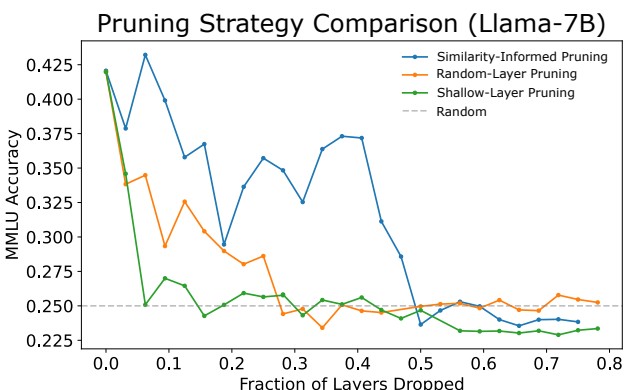

Figure 11: Comparison of the similarity-informed pruning strategy (blue) to random-layer pruning (orange) and shallow-layer pruning (green) on MMLU accuracy, with Llama-2 7B and LoRA rank 64. The similarity-informed pruning strategy clearly outperforms these baselines.