# OpenReview forum: "The Unreasonable Ineffectiveness of the Deeper Layers"
_ICLR.cc/2025/Conference — ICLR 2025 Poster_

### Official Review · Reviewer_9vdV · 2024-10-28

**Soundness:** 2
**Presentation:** 4
**Contribution:** 3
**Rating:** 6
**Confidence:** 3

**Summary:**

This paper conducts a model pruning study, which a) attempts to produce new effective pruning strategies and b) aims to better understand the role of deeper vs. earlier layers in language models.

The pruning method involves removing the deeper layers first (bar the final layer) and show that this method works relatively well, particularly when combined with subsequent training (what is referred to as "healing" in the paper) using QLORA.

The paper presents some ideas suggesting that the later layers may not be as useful, particularly for QA/memorization. Conversely, it shows some preliminary evidence that the deeper layers may be more important for reasoning, albeit these result is left in the Appendix.

**Strengths:**

**Valuable Topic**
The question of how LLM performance degrades as later layers are progressively and systematically pruned is an interesting one, and which requires further exploration.

**Results**
1. The results of the method show relative consistency across model families, albeit some systematic differences are seen between model families, for example in Figure 2, the models undergo "collapse" at significantly different Fraction of layers.

2. The result is shown on a large and robust benchmark, like MMLU.

**Presentation**
Very nice, well-thought out figures which do well in communicating the message.

**Weaknesses:**

**Ablation Study**
I think some key ablation studies have not been undertaken, particularly using control pruning techniques like random pruning, or pruning earlier vs. later layers.

Without these, I find that the claim that the "later layers" are ineffective by itself unconvincing. For example, it could be that removing an equal proportion of layers from randomly chosen locations yield the same results.

**Unclear Conclusions**
While I appreciate the authors efforts in motivating the work, and proposing some interpretation of their results, I found some of the deductions to be confusing or rushed. For example, one of the potentially most interesting results, the difference in behavior between reasoning and QA benchmarks, is left for the Appendix (GSM-8k), with only 1 model, and only fleetingly mentioned in the main paper.

The combination of incomplete ablations, and fleeting/conflated analysis conclusions makes it difficult to take any definitive take-home messages from the paper.

**Questions:**

1. How can we be sure that random pruning or other similar method is not as effective as the
"later layers" approach?
2. How convincing is the evidence about reasoning vs. memorization compared to layer number?
3. Why are results in the appendix selectively shown on differing models? (for example Figure 7, Llama-70b, Figure 8, Llama-13b)

---

### Official Review · Reviewer_1i1x · 2024-11-03

**Soundness:** 3
**Presentation:** 3
**Contribution:** 3
**Rating:** 8
**Confidence:** 4

**Summary:**

The paper shows that deeper LLM layers can often be pruned without significant performance loss in standard QA benchmarks.
The authors propose a pruning strategy to eliminate deeper layers based on the angular distance between the last token representations of nearby layers. The authors also show that if the pruned model is fine-tuned after pruning, it is possible to apply a more straightforward pruning strategy and remove up to 40% of the last blocks without measuring the layer similarity.

**Strengths:**

1. The problem setup is well-defined, and the writing is clear;

2. The claims are supported by experiments on a wide set of models;

3. The findings are interesting, novel, and potentially useful in practice to reduce the memory footprint of LLMs

**Weaknesses:**

The main weakness is the relatively narrow set of downstream benchmark evaluations. The paper's main message relies on multiple-choice QA benchmarks where a single token has to be generated.

The authors briefly discuss the case of GSM8K, showing that it is not always convenient to prune the deeper layers.
Unfortunately, that discussion is briefly mentioned in line 357, and the plot is only shown in the appendix.

I encourage the authors to expand the debate on the limits of applicability of the layer pruning in the main text. Discussing this limitation would make the paper stronger and its contribution to the community more transparent.

**Questions:**

Echoing what I wrote above in the Weaknesses section, I was wondering if the authors have tried open-ended generation tasks beyond GSM8k, such as text summarization. Is it possible to tell if the deeper layers are useful for reasoning or if the pruning hurts any open-ended generation beyond a single token?

---

### Official Review · Reviewer_uhTK · 2024-11-09

**Soundness:** 4
**Presentation:** 4
**Contribution:** 2
**Rating:** 6
**Confidence:** 3

**Summary:**

The authors propose a pruning method for LLMs and conduct studies on LLMs using this approach. Their pruning method involves removing consecutive layers (block) from the LLM, with optional fine-tuning. To determine the size and position of the layer blocks to remove, they suggest either using the similarity of representations or, more simply, removing a specified number of layers starting from the second-to-last layer. Their findings indicate that the deeper layers of the LLM (those closer to the output head) exhibit more similar representations, making them easier to remove.

**Strengths:**

* The paper is clearly written and well-organized, making it easy to follow. The flow of the text, the intuitiveness of the figures, the comprehensive appendix, and their placement all contribute to its readability.
* Despite the simplicity of the proposed pruning method, it achieves substantial gains. Given that the approach is implemented in a discrete, sparse and straightforward manner, it could also serve as an effective tool for understanding LLMs.

**Weaknesses:**

* The paper’s title and claims might somewhat overemphasize the findings, which could lead some readers to interpret them as suggesting that 'LLMs don’t require deep layers'. The experiments are primarily focused on relatively simple QA tasks, while the implications of results from more complex reasoning tasks (like GSM8K) aren’t highlighted enough, despite their significance.

* This highlights a related concern as above: given that one of their contributions is a framework for understanding open-weight LLMs, the range of tasks explored feels limited — mainly knowledge-intensive tasks. To fully clarify the necessity and role of deeper layers in LLMs, testing on more diverse data complexities—tasks requiring higher cognitive abilities, such as reasoning and planning—would be beneficial.

**Questions:**

* I wonder if the authors could present results from a more diverse set of tasks beyond QA to see if the overall trend observed in this paper holds and to explore any potential differences and lessons.
* It would be helpful if the authors provide some quantitative results on the efficiency of the pruning method itself compared to the most similar or representative methodologies.
* This paper presents some interesting findings; however, could the authors offer more practical implications or lessons for improved LLM pre-training schemes beyond simply achieving improved downstream efficiency through pruning?
* Regarding the C4 results, it might be useful for readers to see some qualitative generation outcomes, as this could reveal whether the point at which the semantic validity of the generated sentences begins to collapse aligns with the phase transition observed in the QA results.

**Details Of Ethics Concerns:**

I have reviewed the paper and do not have any specific ethics concerns regarding its content or methodology.

---

### Official Review · Reviewer_TcNy · 2024-11-11

**Soundness:** 3
**Presentation:** 3
**Contribution:** 2
**Rating:** 6
**Confidence:** 4

**Summary:**

This study explores how knowledge is stored in large language models (LLMs) by examining the effects of layer pruning. The research identifies layers that can be removed without impacting the model's performance on knowledge-based benchmarks, suggesting these layers may not be essential for storing knowledge.


The key findings of this work include that: 1) shallow layers may be crucial for knowledge retention, as models are resilient only to pruning of deeper layers; 2) layer pruning, combined with parameter-efficient fine-tuning (PEFT) methods like quantization and Low Rank Adapters (QLoRA), significantly reduces memory use and inference time, with performance scaling well on a single A100 GPU.


Overall, these findings suggest that current pretraining strategies may underutilize deeper layers, and that layer pruning could enhance both the efficiency and accessibility of LLMs in practical applications.

**Strengths:**

The work is written with a great focus on clarity, and the main findings are sufficiently supported.


The main observation, that layer pruning does not impact the performance of world knowledge and comprehension benchmarks, is relevant to the community.

The observation that comparing layers with very intuitive similarity metrics suggests simple and effective layer-pruning strategies is valuable.

**Weaknesses:**

The paper would benefit from further justification that the authors' clear and simple strategies lead to compatible results to possible more sophisticated techniques that emerged in previous literature (see the Questions section (a) for explicit examples).

Several recent efforts approached the study of LLMs representations through geometry. This work would benefit from a more thorough comparison/connection to recent progress in the community along these directions (see the Questions section (b) for explicit points).

**Questions:**

(a) Comparison to other possible comparison and pruning strategies:

- it would be useful to compare the cosine similarity approach adopted by the authors to compare layers to other similarity metrics that appeared for layer comparison (e.g. CKA - Kornblith et al., ICML, 2019, SVCCA - Raghu et al., NIPS, 2017);

- it would be useful to compare strategies that prune successive layers with other possible methods that consider pruning non-consecutive layers (e.g. greedy iterative approach of layer-pruning).

(b) How do the results of the work, and in particular the analyses of Section 4.3, relate to recent findings on the geometry of representations in LLMs found using metrics such as intrinsic dimension (e.g. Valeriani et al., NIPS, 2023; and Cheng et al., ACL, 2023) and linearity (e.g. Park et al., NIPS, 2023; Engels et al., 2024).

(c) It would be beneficial to expand Section 4.2 on the role of the deeper layers by including and discussing more in detail Fig. 7 related to the GSM8k dataset experiment.

---

### Meta-Review · Area_Chair_NmLr · 2024-12-23

**Metareview:**

The paper demonstrates that deeper LLM layers can be pruned without affecting performance on knowledge QA tasks like MMLU, though this doesn't hold for reasoning tasks like GSM-8k. Using layer similarity metrics, they show up to 50% of layers can be removed while maintaining QA performance. Strengths include clear methodology, strong empirical validation across model scales, and practical efficiency implications. Weaknesses include initially narrow task evaluation and limited exploration of alternative pruning approaches. Key gaps are deeper analysis of the reasoning vs. knowledge task differences and broader architectural implications. The paper merits acceptance for providing novel insights about knowledge storage in LLMs and a practical compression method, while thoroughly addressing reviewer feedback through expanded evaluations and ablation studies.

**Additional Comments On Reviewer Discussion:**

The rebuttal addressed major concerns effectively: authors expanded beyond knowledge QA tasks by adding evaluations across HellaSwag, GSM-8k, and QMSum with a new section on task-dependent effects; added ablation studies comparing against random/early-layer pruning baselines; addressed alternative similarity metrics concerns by clarifying their focus on pruning viability; and better scoped their claims to knowledge QA tasks. While compute limitations prevented some comparisons, the comprehensive response with expanded evaluations and new ablations justifies acceptance.

---

### Decision · Program_Chairs · 2025-01-22

Accept (Poster)